# Shedding Light on Osteosarcoma Cell Differentiation: Impact on Biomineralization and Mitochondria Morphology

**DOI:** 10.3390/ijms24108559

**Published:** 2023-05-10

**Authors:** Francesca Rossi, Giovanna Picone, Concettina Cappadone, Andrea Sorrentino, Marta Columbaro, Giovanna Farruggia, Emilio Catelli, Giorgia Sciutto, Silvia Prati, Robert Oliete, Alice Pasini, Eva Pereiro, Stefano Iotti, Emil Malucelli

**Affiliations:** 1Department of Pharmacy and Biotechnology, University of Bologna, 40126 Bologna, Italy; francesca.rossi105@unibo.it (F.R.); giovanna.picone2@unibo.it (G.P.); concettina.cappadone@unibo.it (C.C.); giovanna.farruggia@unibo.it (G.F.); emil.malucelli@unibo.it (E.M.); 2Mistral Beamline, ALBA Synchrotron Light Source, Cerdanyola del Valles, 08290 Barcelona, Spain; asorrentino@cells.es (A.S.); roliete@cells.es (R.O.); epereiro@cells.es (E.P.); 3Piattaforma di Microscopia Elettronica, IRCCS Istituto Ortopedico Rizzoli, 40136 Bologna, Italy; marta.columbaro@ior.it; 4National Institute of Biostructures and Biosystems (NIBB), 00136 Rome, Italy; 5Department of Chemistry “G. Ciamician”, Università di Bologna, Ravenna Campus, Via Guaccimanni, 42, 48121 Ravenna, Italy; emilio.catelli2@unibo.it (E.C.); giorgia.sciutto@unibo.it (G.S.); s.prati@unibo.it (S.P.); 6Department of Electrical, Electronic and Information Engineering “Guglielmo Marconi” (DEI), University of Bologna, Via dell’Università 50, 47522 Cesena, Italy; alice.pasini@unibo.it

**Keywords:** osteogenic sarcoma, bone cancer, osteoblastic differentiation, biomineralization, mitochondria, cryo-XANES, calcium L-edge

## Abstract

Osteosarcoma (OS) is the most common primary malignant bone tumor and its etiology has recently been associated with osteogenic differentiation dysfunctions. OS cells keep a capacity for uncontrolled proliferation showing a phenotype similar to undifferentiated osteoprogenitors with abnormal biomineralization. Within this context, both conventional and X-ray synchrotron-based techniques have been exploited to deeply characterize the genesis and evolution of mineral depositions in a human OS cell line (SaOS-2) exposed to an osteogenic cocktail for 4 and 10 days. A partial restoration of the physiological biomineralization, culminating with the formation of hydroxyapatite, was observed at 10 days after treatment together with a mitochondria-driven mechanism for calcium transportation within the cell. Interestingly, during differentiation, mitochondria showed a change in morphology from elongated to rounded, indicating a metabolic reprogramming of OS cells possibly linked to an increase in glycolysis contribution to energy metabolism. These findings add a dowel to the genesis of OS giving new insights on the development of therapeutic strategies able to restore the physiological mineralization in OS cells.

## 1. Introduction

Osteosarcoma (OS) is the most common primary malignant bone cancer affecting children and young adults [1]. These tumors usually occur in the metaphysis of long bones where high turnover occurs. The current standard treatment for OS involves neoadjuvant chemotherapy and surgical resection of the tumoral tissue [2,3]. Indeed, both therapeutic approaches face numerous problems such as chemoresistance, treatment adverse effects and adequate surgical margins [4,5,6]. Although its origins have not been fully elucidated, several investigations have suggested osteogenic differentiation dysfunctions as key factors in OS etiology [2,7]. During osteoblastic differentiation, affected cells are likely to arrest the process at the undifferentiated precursors stage and produce malignant bone tissue as a result of bone biomineralization [5,8,9]. Hence, OS cells respond to growth factors in the same manner as early osteoprogenitors and proliferate, leading to cancer progression. Furthermore, it has been observed that more malignant OS phenotypes share similarities with early osteogenic progenitors, while less aggressive ones resemble more differentiated bone mesenchymal stem cells (bMSCs) [2,4,7]. Cell differentiation is an energy-demanding process leading to metabolic changes associated with mitochondrial morphology variations and it is well known that changes in mitochondria shape are related to cell functions through fission and fusion processes [10,11,12]. However, the role of mitochondria in differentiation and biomineralization has not been completely elucidated. In this context, a deeper understanding of differentiation and biomineralization in both OS cells and primitive osteoblasts becomes crucial for the development of new effective treatments.

Biomineralization is fundamental for bone formation, repair and modelling, and culminates in the growth of a complex tissue composed of hydroxyapatite crystals and collagen fibers [13]. However, the pathways leading to the generation of intracellular crystalline materials have been poorly characterized so far. Marchegiani and colleagues have described in vitro crystalline hydroxyapatite (HA) formation as an evolution from amorphous calcium carbonate to calcium phosphate compounds [14]. A similar process has recently been observed in bMSCs [15]. Specifically, in differentiating bMSCs, biomineralization starts intracellularly with the nucleation of calcite that evolves towards hexagonal HA crystals, which are the main mineral components of mature human bone [15,16]. In human OS cells, the genesis and evolution of intracellular mineral depositions need to be further elucidated. Recent studies on bone cell differentiation have suggested that mitochondria act as calcium vehicles during osteogenic commitment. Therefore, these organelles are actively involved in intracellular calcium trafficking during the biomineralization process. Calcium deposits are likely to be transferred from the endoplasmic reticulum (ER) to mitochondria and then transported to intracellular vesicles that vehiculate Ca towards the collagenous extracellular matrix [17,18].

In this study, we induced the osteoblast-like human osteosarcoma cell line SaOS-2 to differentiate into a mature osteoblastic phenotype, which was compared to OS cells used as reference. The early stages of bone biomineralization were studied by monitoring the genesis of the first mineral core deposits and their evolution during the differentiation process using gene expression analysis, histochemical staining, enzyme activity assays and transmission electron microscopy. Moreover, synchrotron-based cryo-soft X-ray tomography (SXT) and cryo-XANES spectro-microscopy were employed to follow the chemical evolution of mineral depositions over time and localize them in the cellular ultrastructure, allowing to simultaneously track morphological variations in intracellular organelles at nanometric resolution.

## 2. Results

### 2.1. Gene Expression Analysis and Functional Characterization of SaOS-2 Cells after Osteogenic Induction

After seven days of induction, red granules were evident in both the control and treated samples, but the latter exhibited more intensely marked calcified nodules (Figure 1A). In addition, a gene expression analysis of six osteogenic markers was performed (Figure 1B). Relevant upregulation of these gene transcripts was observed in the treated samples, except for osteonectin (SPARC1) mRNA. In particular, the master osteogenic transcription factor (RUNX2) and its downstream controlled gene alkaline phosphatase (ALPL) showed a significant increase in mRNA expression levels. The COL1A1 gene transcript was markedly increased, reaching a nine-fold higher expression when compared with control cells. A five-fold upregulation was also observed in the late osteogenic markers osteocalcin (BGLAP) and osteopontin (SPP1). Since the ALPL enzyme plays a key role in bone mineralization and osteogenic differentiation, we assayed its catalytic activity and scored a positive correlation with the observed increased gene expression (Figure 1C). It is worth noting that osteogenic stimulation led to a relevant increase in SaOS-2 cells alkaline phosphatase activity despite its already high baseline levels [19]. In addition, ultrastructural analysis by transmission electron microscopy (TEM) confirmed the osteogenic commitment after seven days of induction, showing a mineralization front, characterized by the deposition of inorganic crystals on aggregated collagen fibrils outside the cytoplasmic membrane (Figure 1D, red arrow). Numerous vesicles were also observed, some of which could be ascribed to matrix vesicles (Figure 1D, red asterisks) located at the sites of mineralization. In control SaOS-2 cells, no mineralized areas were evident. Overall, overexpression of early (RUNX, COL1A1, ALPL) and late (BGLAP, SPP1) osteogenic genes, increased ALPL activity and the presence of extracellular mineral matrix underlined the osteogenic commitment during differentiation.

### 2.2. Characterization of Ca-Depositions by Cryo-XANES and Cryo-SXT

The combination of cryo-soft X-ray tomography (SXT) with cryo-XANES imaging is a well-established strategy that allows characterizing of the spatial distribution and the chemical state of intracellular structures at the nanoscale [20]. The early steps of bone mineralization in SaOS-2 cells were investigated by combining these techniques at the Ca L_2,3_ edges. SaOS-2 at two different induction times, 4 days (4D) and 10 days (10D), were grown on electron microscopy grids, plunge frozen in liquid ethane and were subsequently imaged in a quasi-native state (unfixed, unstained and unsectioned) using the soft X-ray transmission microscope installed at the Mistral beamline of the Alba synchrotron [21].

#### 2.2.1. Ca Distribution and Chemical Fingerprint of Ca Depositions

The chemical composition of intracellular Ca depositions was characterized by cryo-XANES spectro-microscopy, providing structural information about the atoms surrounding the first coordination sphere of the Ca and its spatial 2D distribution at the nanoscale. A typical Ca L_2,3_ edge spectrum presents two main spin-orbit related peaks (L_3_ ≈ 349.2 eV; L_2_ ≈ 352.9 eV) and a “multi-peak pattern” of relatively lower intensity before them. The region of the spectrum with energy below 345 eV, named the pre-edge, does not show any Ca absorption [22]. Thus, to specifically locate Ca depositions within the cell, the energy of the incident beam was varied across the Ca L_2,3_ edges. Figure 2 shows the transmission images of control and differentiated SaOS-2 cells (4D and 10D) at the Ca pre-edge (Figure 2A–C) and at the L_3_ peak maxima (Figure 2D–F), allowing identifying of the Ca depositions (green arrows in Figure 2D–F). At 4 days in the SaOS-2s control, small Ca depositions were detected (Figure 2D light green arrows), while in treated SaOS-2 an increase in the size and number of Ca depositions was found (Figure 2E, green arrows) indicating early osteogenic commitment. After 10 days of osteogenic differentiation, Ca depositions increased in number even more and evolved into more compact and dense structures (Figure 2F, dark green arrows).

Ca deposition in the same cell showed very similar XANES spectra. Average XANES spectra from control SaOS-2 at 4 days (Figure 2G) showed intense L_3_ and L_2_ pre-peaks matching the calcite (CaCO_3_) reference spectrum (a_1_ and b_1_ in Figure 2J). It is worth noting that at 4 days the differentiating SaOS-2 generated depositions with spectral profiles (Figure 2H) similar to that of the calcium phosphate reference (Ca_3_(PO_4_)_2_), as reported in Figure 2K, suggesting a change in the deposition chemical state and a timeline of biomineralization. After 10 days of osteogenic differentiation, the crystalline structure of the mineral depositions further evolved towards hydroxyapatite (HA) (Figure 2I), as inferred from the comparison with the HA reference spectrum (Figure 2L). In fact, both spectra exhibit the main characteristic L_3_ and L_2_ peaks of calcium compounds and the low-intensity peaks structure before the L_3_ main peak (a_1_, a_0_ in Figure 2L). In addition, in reference and cell spectra the “hook” shape of the L_2_ pre-peak (b_1_ in Figure 2L), which is typical of the HA Ca-L edge absorption spectrum, can be easily observed [23].

For a more in-depth description of the spectroscopic data, we used principal component analysis (PCA) to better understand whether a distinction exists among the XANES spectra at 4 and 10 days and whether the separation may reflect the observed chemical changes. It is worth noting that PCA takes into consideration the whole spectral profile (Figure 3A,C), enhancing important information such as peaks shape and intensity.

Prior to the analysis of the cells, we applied PCA to standard spectra of calcium phosphate Ca_3_(PO_4_)_2_ and HA, to evaluate the efficacy of the data processing in the extraction of the relevant information from XANES results (Figure 3A). The score PC1 vs. PC2 plot (Figure 3B) showed an efficient discrimination between calcium phosphate and HA, thanks to their peculiar spectral features within the entire energy region of interest (between 347 and 355 eV). Concerning the PCA results related to the spectra acquired from cells (Figure 3C,D), it is possible to observe a good separation between the 4D and 10D spectra, confirming the evolution of the chemical composition of Ca depositions. In this case, the discrimination seems to be enhanced by changes in shape and relative intensity presented by the L_3_ (at ≈349 eV) and L_2_ (at ≈353 eV) main peaks, by the a_0_ pre-peak, previously mentioned as a discriminant between calcium phosphate and HA and the b_1_ pre-peak.

#### 2.2.2. 3D Cell Ultrastructure Analysis

The extension to 3D is obtained by combining cryo-XANES imaging with cryo-SXT, which allows a reconstruction of the cellular volume rich in structural details and enables a nanoscale intracellular location of Ca depositions. Projections at different angles (see Section 4) were acquired using a photon energy of 352.9 eV at which the Ca-rich structures show higher absorption contrast. In the following, the main morphological features observed in the SaOS-2 samples are illustrated.

Tomograms from 4-day control samples showed elongated mitochondria and a few small CaCO_3_ depositions in the cellular environment. (Figure 4A,B). SaOS-2 cells at 4 days after induction provided a quite different scenario: elongated mitochondria consistently decreased in number, leaving room for round mitochondria containing highly absorbing Ca structures (Figure 4C) as highlighted by the transmission images in Appendix A. Round mitochondria appear grouped (Figure 4D) suggesting the occurrence of a fission process. Indeed, it is well known that mitochondria change their morphology under different physiological conditions, playing a pivotal role in calcium transportation and depositions genesis and evolution within the cell [24].

A magnified view of one of these round mitochondria is shown in Figure 4E and the corresponding 3D rendering (Figure 4F) highlights the ordered and structured organization of Ca, as verified by the XANES spectrum in Figure 4G. These spectra (Figure 4G) can be attributed to some form of Ca carbonate, which is different from that of calcite [25]. Interestingly, round mitochondria appeared in close contact with vesicles (Figure 5A, blue arrow) containing calcium phosphate depositions (Figure 5A, green arrow) and calcium transfer among them could be observed (Figure 5A, pink arrow). Figure 5B provides the 3D rendering of the reconstruction of Figure 5A.

After 10 days of osteogenic differentiation, the scenario further evolved with respect to 4 days. Round mitochondria started changing their shape and became more elongated and bound (Figure 5C,D), mimicking a mitochondrial fusion process. In addition, mitochondria appeared to be linked together, suggesting the formation of a mitochondrial network (Figure 5E) and the restoration of the conventional mitochondria shape. In the reconstruction slices reported in Figure 5C,D, these organelles still exhibited highly absorbing structures; however, in this case, they were not made of Ca (see Appendix A). This can be a consequence of a possible Ca transfer to depositions during the early days of differentiation as described above and highlighted by Figure 5A,B.

## 3. Discussion

A deeper understanding of the relationship between defects in osteogenic differentiation and malignant bone tissue formation could help the comprehension of osteosarcoma tumorigenesis and the development of new treatment strategies. Several studies have pointed out that OS might be associated with differentiation defects in mesenchymal stem cells [2], leading to abnormal biomineralization. In order to characterize the defective biomineralization process that occurs in OS, we studied osteoblast-like SaOS-2 cells during the early stages of differentiation, monitoring the genesis and evolution of the bone mineral matrix. At 4 days after osteogenic induction, SaOS-2 cells produced calcium phosphate depositions that evolved towards hexagonal HA crystals up to day 10 of differentiation (Figure 2, 2nd and 3rd columns). The discrimination of calcium phosphate (Ca_3_(PO_4_)_2_) and hydroxyapatite (Ca_5_(PO_4_)_3_(OH)) chemical structures was achieved by PCA of the XANES spectra acquired from the mineral depositions. In contrast, untreated 4-day SaOS-2 cells exhibited mineral depositions mainly composed of calcite (CaCO_3_) (Figure 2, 1st column). The presence of calcite in the control samples confirms that early differentiated osteoprogenitors derived from bMSCs and SaOS-2 cells are similar, supporting the hypothesis that OS cells arise from MSCs unable to undergo complete differentiation [2,3,7]. The evolution of mineral deposits from calcium phosphate to HA induced by the differentiating treatment suggests a restoration of the physiological biomineralization process in SaOS-2 cells, as previously assessed in differentiate bMSCs [15].

Recently, Boonrungsiman et al. identified calcium phosphate granules in mouse osteoblasts within mitochondria and intracellular vesicles. They observed that Ca-containing mitochondria conjoined with vesicles, confirming the role of mitochondria as a calcium storehouse and suggesting a transport mechanism driven by vesicles [17]. In the present study, similar features were observed in SaOS-2 human cells. Small Ca structures were spotted in mitochondria at 4 days from osteogenic induction (Figure 4F). At this point in the differentiation timeline, some mitochondria were linked to vesicles containing calcium phosphate depositions highlighting calcium transfer between the two intracellular organelles (Figure 5A). At 10 days after induction, no Ca minerals were detected in the mitochondria, supporting the interplay between mitochondria and vesicles in calcium trafficking (Figure 5C). A complete scheme of the possible mechanism of Ca nucleation and transportation during biomineralization is provided in Figure 6.

In addition, the mitochondrial morphology was monitored using cryo-SXT during SaOS-2 cells differentiation. Despite being cancer cells, untreated SaOS-2 cells exhibited elongated mitochondria (Figure 4A), a feature mainly associated with increased oxidative phosphorylation (OXPHOS) [10]. In fact, SaOS-2, defined as osteoblast-like cells, are characterized by a partial differentiation with respect to other OS cell lines that are less prone to osteogenesis [26,27]. Indeed, SaOS-2 cells expressed typical genes of mature osteoblasts, such as ALPL, BGLAP and SPP1 (Figure 1B). Interestingly, non-fused and fragmented mitochondria were observed four days after induction (Figure 4E), suggesting an increase in glycolysis contribution to energy metabolism [10,28]. Instead, at 10 days after induction, fused and interconnected mitochondria were observed (Figure 5D,E).

In conclusion, this study sheds light on the genesis and evolution of HA in differentiating osteosarcoma cells, identifying HA as the final product of the restored biomineralization. Furthermore, our findings suggest a mechanism for mineral matrix transport based on the interaction between mitochondria and vesicles. Finally, through a morphological study of mitochondria, reprogramming of OS cells induced to differentiate was observed, as similarly reported for other cancer types [29,30,31]. Nevertheless, further investigations on other OS cell lines are necessary to corroborate the mitochondria-mediated mechanism indicated by our study.

These findings reinforce the theory that osteosarcoma originates from defective osteoblastic differentiation, supporting therapeutic approaches based on differentiating agents that might restore physiological mineralization in OS cells.

## 4. Materials and Methods

All reagents were purchased from Sigma-Aldrich (St. Louis, MO, USA) unless otherwise specified.

### 4.1. SaOS-2 Cell Culture and Osteogenic Induction

Human osteosarcoma cell line SaOS-2 cells were purchased from American Type Culture Collection (ATCC, Manassas, VA, USA). Cells were grown in RPMI 1640, supplemented with 10% heat inactivated FBS, 2 mM glutamine, 1000 units/mL penicillin and 1 mg/mL streptomycin, at 37 °C in a 5% CO_2_ 95% air humidified atmosphere. After 24 h, cells were treated with vehicle or osteogenic differentiation cocktail containing 20 nM 1,25-Dihydroxyvitamin D3, 50 µM L-Ascorbic acid 2-phosphate and 10 mM -glycerol phosphate, replacing the media every 48 h.

### 4.2. Alizarin Red Staining

SaOS-2 cells were seeded at 1 × 10^4^/cm^2^ and treated with an osteogenic cocktail after 24 h. Furthermore, 2% alizarin red S was dissolved in distilled water and the pH was adjusted to 4.1–4.3 using 0.5% ammonium hydroxide. Cultures were fixed with cold EtOH 70% for 30 min at room temperature, washed and stained with alizarin red S solution for 30 min. After removal of unincorporated excess dye with distilled water, the mineralized nodules were labelled as red spots.

### 4.3. Alkaline Phosphatase Activity Assay

After 7 days of treatment, the cells were freshly harvested, washed twice in ice-cold phosphate buffered saline (PBS) and centrifuged at 3000 rpm for 5 min. For the treated samples, a cell scraper was used to detach cells and collect them into ice-cold PBS before pelleting via centrifugation. Cells were lysed in ice by a 1 mL solution of 40 mmol/L HEPES, 110 mmol/L NaCl, 0.25% deoxycholate and 1 mg/mL aprotinin, pH 7.4. The homogenate was used for the ALP spectrophotometric assay by using *p*-nitrophenyl phosphate as a substrate. ALP activity was normalized for the protein content. Proteins were measured by the Bio-Rad protein assay method. One unit of ALP activity is defined as the amount of protein capable of transforming 1 mmol of substrate in 1 min at 25 °C. Statistical analysis was performed using GraphPad Prism 8 software using a *t*-test and differences were deemed significant for * *p* < 0.05.

### 4.4. Gene Expression Analysis

Total RNA from cells was collected after 7 days of osteogenic induction and extracted using the NucleoSpin RNA (Macherey Nagel, Düren, Germany) following the manufacturer’s instructions. The level of expression of the osteogenic markers runt-related transcription factor 2 (RUNX2), collagen type I (COL1A1), osteocalcin (BGLAP), osteopontin (SPP1) and osteonectin (SPARC) was analyzed by quantitative real-time PCR (qPCR), as previously reported [32]. Glyceraldehyde 3-phosphate dehydrogenase (GAPDH) and hypoxanthine phosphoribosyltransferase 1 (HPRT1) were used as housekeeping reference genes (2^−∆∆CT^ method). Primer sequences are reported in Table 1. Fold changes from untreated control cells were calculated. Data analysis was performed using CFX Manager Software (Bio-Rad), creating a gene study that uses an internal calibrator to normalize the variability between the experiments. Data are reported as mean value ± SEM of at least three independent biological replicates. Statistical analysis was performed using GraphPad Prism 6 software using a two-way ANOVA followed by a Sidak’s multiple comparison test. *p* values less than 0.05 were accepted as significant.

### 4.5. Transmission Electron Microscopy (TEM)

SaOS-2 cells was seeded in 6-well plates at 1 × 10^4^ cells/cm^2^ and were treated with vehicle or osteogenic differentiation cocktail as previously reported. After 4, 7 and 10 days of osteogenic treatment, the samples were fixed with glutaraldehyde (2.5%) in cacodylate buffer (0.1 M pH 7.4) for 10 min and then detached using a cell scraper. The cells were harvested, centrifuged at 3000 rpm for 20 min and fixed for 50 min at RT in the same solution. Post fixation, the pellets were washed twice in 0.1M cacodylate buffer of pH 7.6. After post-fixation with 1% osmium tetroxide (OsO4) in 0.1 M sodium cacodylate buffer for 1 h, the pellets were dehydrated in an ethanol series, infiltrated with propylene oxide, and embedded in Epon resin. Ultrathin sections (80 nm thick) were stained with uranyl acetate and lead citrate (15 min each) and were observed with a JEOL JEM-1011 transmission electron microscope, operated at 100 kV. At least 100 cells per sample were observed.

### 4.6. X-Ray Absorption Near-Edge Spectro-Microscopy (XANES-Imaging) and Cryo-Soft-X-Ray Tomography (Cryo-SXT)

SaOS-2 cells were seeded onto gold QUANTIFOIL R 2/2 holey carbon-film microscopy grids. The cells were plated at a concentration of 2 × 10^4^ cell/cm^2^ on the grids previously sterilized by UV light for 3 h. After 4 and 10 days from osteogenic induction, the attachment and spreading of the cells was carefully verified using visible light microscopy prior to freezing. Immediately prior to freezing, 1.5 µL of 100 nm gold fiducial aliquot (chemical reference: gold nanoparticles 100 nm, EMGC100, BBI Group, Cardiff, UK) was added to the grids, necessary as external marker for the alignment needed for tomographic reconstruction. The grids were frozen hydrated by rapid plunge freezing in a liquid ethane pool cooled with liquid nitrogen using a GP-Leica Microsystems Excess. Before plunge freezing, water is removed via blotting. Quality of the sample’s preparation was checked in cryo-conditions prior to loading into the microscope chamber. After plunge freezing, cryogenic temperature (~110 K) is maintained until and during the measurements. Calcite (Sigma-Aldrich), HA and Ca_3_(PO_4_)_2_ (Bio Eco Active S.R.L, Bologna, Italy) references were prepared by crushing powders in a mortar, and the obtained dust was laid down on a Quantifoil Au TEM grid.

Cryo-SXT and cryo-XANES images were recorded at the Mistral beamline of the ALBA Synchrotron. Scanning the X-ray energy through the Ca L_3,2_ edge, cryo-XANES can be used to determine the Ca bulk chemical state [33]. Two-dimensional Ca L_3,2_ edge XANES images (12 s exposure time) were collected on selected areas using an effective pixel size of 13 nm and with a variable energy step (0.5 eV of pre-edge and post-edge, 0.1 elsewhere). The total acquisition time was about 1.5 h per energy scan, including the flat field acquisition at each energy step.

Two-dimensional XANES images were processed and derived according to Sorrentino et al. [15]. In particular, straylight background, mainly due to the 1st negative zone plate order, was estimated following the procedure described in [34] on the calcite reference sample. Pixels with measured transmission smaller than the background were filtered out, again using a homemade function created in Matlab.

Cryo-SXT was carried out at 352.9 eV to optimize the contrast between the calcium- and carbon-rich structures and the surrounding water-rich cytoplasmic solution. For each cell, a tilt series was acquired using an angular step of 1° on a 110° angular range with effective pixel size equal to 13 nm at 352.9 eV. Each image of the tilt series was normalized using flat-field images of 1 s acquisition time. The tilt series were manually aligned using eTomo in the IMOD tomography software suite [35] using Au fiducials of 100 nm. The aligned tilt series were reconstructed using TOMO3D software, using the SIRT algorithm (30 iterations), and then segmented by Amira (Thermo Fisher Scientific, Waltham, MA, USA).

### 4.7. Principal Components Analysis (PCA)

PCA is one of the most efficient and common tools for multivariate data exploration and reduction. By representing the system through a new uncorrelated set of orthogonal variables obtained as linear combinations of the original variables and called PCs, PCA captures the original data variance and reveals similarity, groupings and trend patterns, that may characterize the objects (spectra), according to their similarities in chemical properties. The new coordinates of the objects are called scores, while the loadings represent the importance of each original variable in defining each PC direction. Score and loading values can be represented on bidimensional scatter plots (score and loading plots, respectively), which allow visualization of the similarities between the objects and the inter-correlation among original variables.

PCA was initially applied on a matrix containing standard spectra (3 spectra of standard calcium phosphate and 22 spectra of standard hydroxyapatite). Afterward, PCA was applied to the matrix of the spectra of the real samples obtained from the Ca-depositions at four days and at ten days for a total of 67 spectra (objects). In both cases, all 85 energies available considered in the XANES spectra collected, were considered as variables.

Prior to PCA analysis, the row spectra were mathematically pre-treated to minimize the unwanted physical effects and noise. The pre-treatment step flow, applied equally to the standard and real samples spectra, consisted of (1) spectral range selection of the main calcium features; (2) range scaling (RS), where the pre-peak and peak regions were scaled with the formula used by Sciutto et al. [36], to have a comparable variability range along the y-axis (intensity); (3) Savitzky–Golay first derivative and (4) column centering.

## Figures and Tables

**Figure 1 ijms-24-08559-f001:**
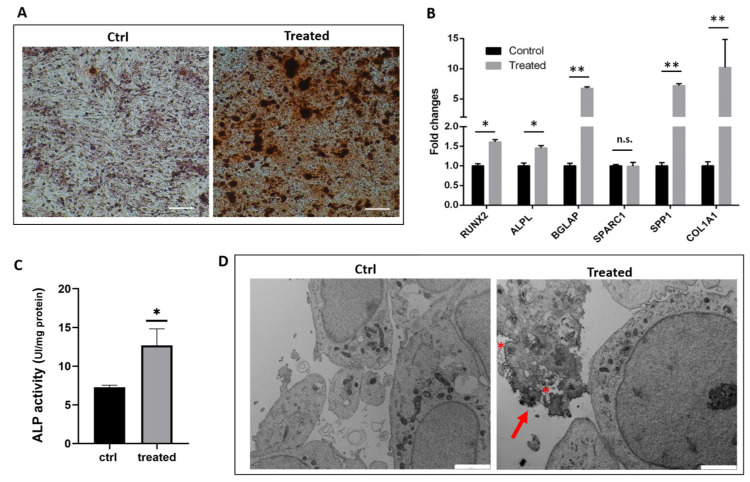
Effects of osteogenic induction on SaOS-2 cell culture at 7 days. (**A**) Alizarin red staining. Scale bar is 200 µm. (**B**) qPCR analysis of osteogenic markers (RUNX2, ALPL, SPARC, COL1A1, BGLAP and SPP1) was performed using GAPDH and HPRT1 as reference genes (2^−∆∆CT^ method). Fold changes from untreated control cells were calculated. Data from three biological replicates are reported as mean ± standard error (SE). Two-way ANOVA followed by Sidak’s multiple comparison test was performed, * *p* < 0.05; ** *p* < 0.01. (**C**) Alkaline phosphatase activity. Data are reported as the mean ± SE of three biological replicates. Unpaired T-test was performed, ** p* < 0.05. (**D**) Representative TEM images of control and treated SaOS-2 cells. In the treated sample, red arrow highlights the biomineralization front outside the cell membrane and asterisks indicated supposed matrix vesicles. Magnification 12,000×. At least 100 cells per sample were observed. Scale bar is 2 µm.

**Figure 2 ijms-24-08559-f002:**
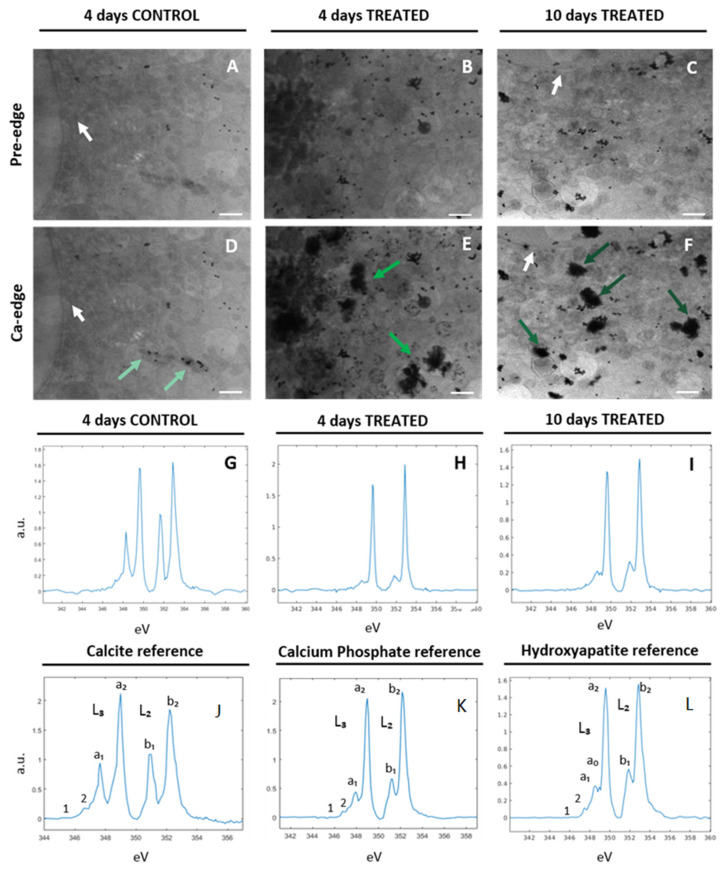
Cryo-XANES spectro-microscopy carried out on control and treated SaOS-2 cells after 4 and 10 days of osteogenic treatment. Average of absorbance projections recorded at the pre-Ca-edge energy region (≈342 eV) and at the Ca L_3_ peak maxima (≈349 eV), for control 4 days (**A**,**D**), treated 4 days (**B**,**E**) and treated 10 days (**C**,**F**), respectively. The contrast between Ca and other elements was maximized to show the presence of calcium depositions. White arrows indicate the edge of cell nuclei, and green arrows indicate the Ca depositions at the L_3_ edge. Scale bar is 1 µm. (**G**,**J**) show the average XANES spectrum of CaCO_3_ extracted from control cells at 4 days and CaCO_3_ reference spectra, respectively. (**H**) represents Ca_3_(PO_4_)_2_ spectrum from a Ca deposition in treated samples at 4 days; (**K**) Ca_3_(PO_4_)_2_ reference spectrum. (**I**,**L**) show representative HA spectra detected in a deposition of SaOS-2 cells after 10 days of osteogenic treatment and HA reference spectra, respectively. In L, M and N the main spectral features are indicated by letters and numbers.

**Figure 3 ijms-24-08559-f003:**
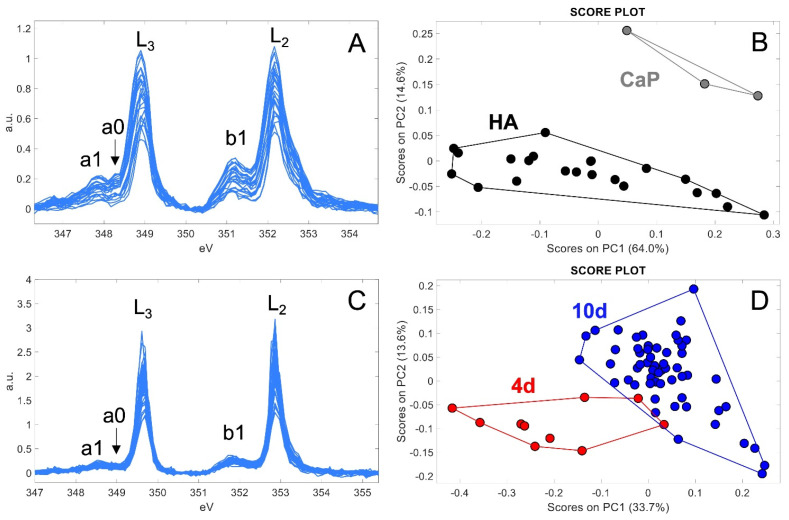
Results of principal component analysis applied to XANES spectra: (**A**) measured standard spectra of calcium phosphate (CaP) and hydroxyapatite (HA); (**B**) score plot showing the separation between calcium phosphate and HA spectra; (**C**) measured spectra from Ca depositions taken at 4D and 10D after the treatment; (**D**) score plot showing the separation between 4D and 10D spectra.

**Figure 4 ijms-24-08559-f004:**
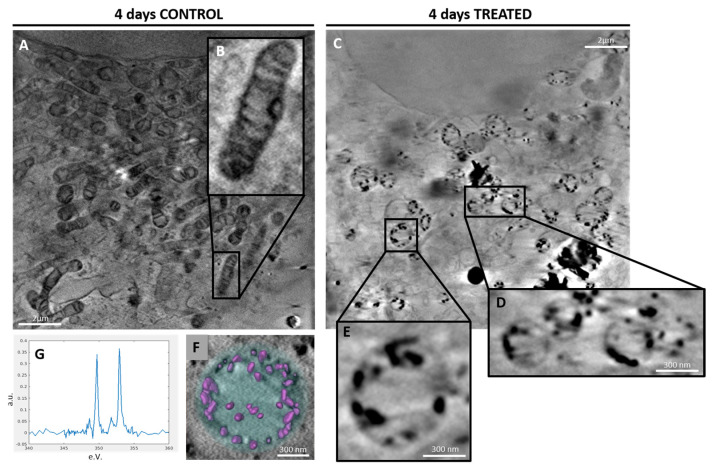
(**A**) Tomogram slice of a 4-day control sample and (**B**) a zoom on a mitochondrion. (**C**) shows a tomogram slice of a 4-day treated sample. (**D**,**E**) provide a zoom of round mitochondria that are close together and a mitochondrion, respectively. (**F**) shows a 3D reconstruction of the mitochondrion in (**E**): Ca structures are highlighted in pink and their corresponding XANES spectrum can be appreciated in (**G**) (see Appendix A).

**Figure 5 ijms-24-08559-f005:**
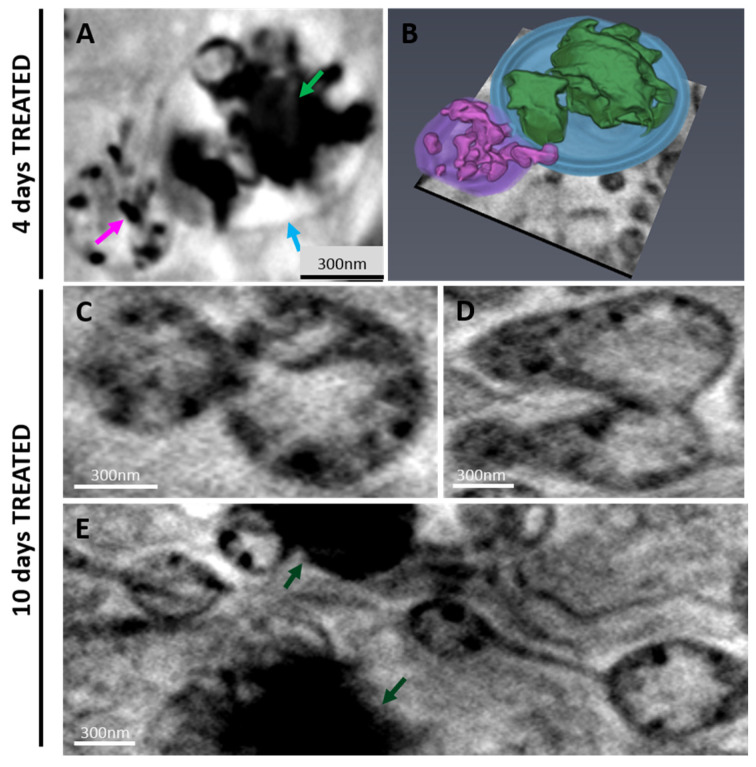
Tomogram slices of a 4D treated sample acquired at 352.9 eV; in (**A**) a zoom on a round mitochondrion transferring Ca (pink arrow) to a calcium phosphate deposition (green arrow) contained within a vesicle (blue arrow). The corresponding 3D reconstruction is provided in (**B**). Tomogram slices of a 10D treated sample acquired at 352.9 eV: (**C**) provides a zoom on mitochondria containing elements different from Ca, (**D**,**E**) show a mitochondrion during a fusion process and a zoom on a mitochondrial network, respectively. Dark green arrows indicate HA depositions.

**Figure 6 ijms-24-08559-f006:**
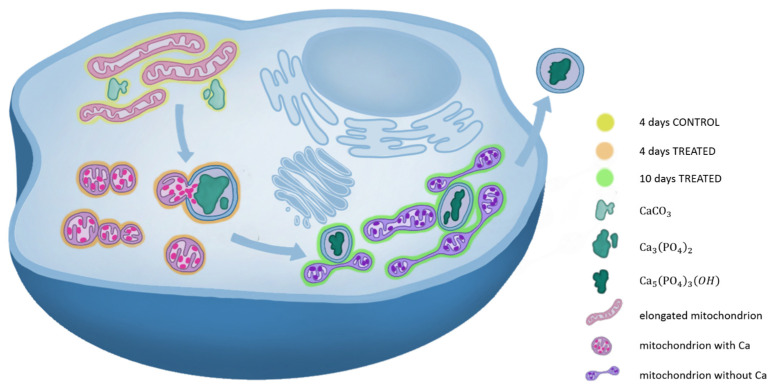
Scheme of an osteosarcoma cell. In yellow, orange and green, are highlighted representative cytoplasmic environments of a 4 days control, 4 days treated and 10 days treated sample, respectively. Calcium depositions are colored in different shades of green. From the lightest to the darkest: calcite, calcium phosphate and hydroxyapatite. Elongated mitochondria can be observed in light pink, mitochondria containing calcium in pink and mitochondria after calcium transfer are in violet.

**Table 1 ijms-24-08559-t001:** Primer sequences used for qPCR.

Gene	Sequence 5′-3′	
GAPDH	F	ACAGTTGCCATGTAGACC
	R	TTGAGCACAGGGTACTTTA
HPRT1	F	ATAAGCCAGACTTTGTTGG
	R	ATAGGACTCCAGATGTTTCC
RUNX2	F	AAGCTTGATGACTCTAAACC
	R	TCTGTAATCTGACTCTGTCC
ALPL	F	TCTTCACATTTGGTGGATAG
	R	ATGGAGATATTCTCTCGTTC
SPARC	F	AGTATGTGTAACAGGAGGAC
	R	AATGTTGCTAGTGTGATTGG
COL1A1	F	CCAGCAAATGTTCCTTTTTG
	R	AAAATTCACAAGTCCCCATC
BGLAP	F	TTCTTTCCTCTTCCCCTTG
	R	CCTCTTCTTGAGTTTATTTGG
SPP1	F	GACCAAGGAAAACTCACTAC
	R	CTGTTTAACTGGTATGGCAC

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
