# Peer review of "Shedding Light on Osteosarcoma Cell Differentiation: Impact on Biomineralization and Mitochondria Morphology"

_ijms, 2023, doi:10.3390/ijms24108559_

Round 1

Reviewer 1 Report

Osteosarcoma (osteogenic sarcoma) is a malignant tumor whose cells form atypical osteoid or bone structures (neoplastic, tumor osteogenesis); one of the most heterogeneous human tumors. Microscopic properties can vary considerably among different tumors and within the same tumor. With the exception of myeloma, it is the most common primary bone tumor (accounting for about 20% of all skeletal sarcomas). Depending on the predominance of the type of tumor cells and the substance produced by them, 3 variants of osteosarcomas are distinguished: osteoblastic - the production of tumor bone or osteoid predominates; chondroblastic - cartilaginous differentiation predominates; fibroblastic - fibroblast-like elements predominate. Osteosarcoma usually develops after the age of 10 years and very rarely before the age of 5 years. The presence of certain factors can increase the risk of developing osteosarcoma. Osteosarcoma is somewhat more common in male patients than in female patients. In cancer survivors who received radiation therapy, the risk also increases. Genetic factors may increase the risk. A small percentage of children have changes or mutations in genes that increase the risk of developing osteosarcoma and other types of cancer.

The authors have done a great job in the study of osteosarcoma. All points of the material are detailed and clear. The authors show the results in tomograms, diagrams. Thus, all information is presented to the reader in a very clear and detailed way. This work is done at a very high scientific level and without a doubt deserves to be published.

Author Response

The authors have done a great job in the study of osteosarcoma. All points of the material are detailed and clear. The authors show the results in tomograms, diagrams. Thus, all information is presented to the reader in a very clear and detailed way. This work is done at a very high scientific level and without a doubt deserves to be published.

Thank you for your kind comments. We appreciate the time and the effort that you have dedicated to providing a valuable feedback on our manuscript.

Reviewer 2 Report

Thank you for giving me the opportunity to review this article. This manuscript is well described, focusing on intracellular Ca metabolism and its relationship with the carcinogenesis of osteosarcoma. The introduction, results, and discussion are also clearly stated. However, some minor modifications and additional explanations are necessary. The followings are specific points.   (Materials and Methods) -In Gene expression analysis, the DNA sequence of each primer should be noted.  -Please describe the internal control and its primer sequence.  (Discussion) All results of this study are based solely on the SaOS-2 cell line. Therefore, I think the evidence is weak in terms of reproducibility. Please mention that as a limitation in the discussion.  (Overall) -Some abbreviations need to be explained when used for the first time. (e.g., HA, TEM, RT, and PBS.)

Author Response

(Materials and Methods) -In Gene expression analysis, the DNA sequence of each primer should be noted.  -Please describe the internal control and its primer sequence.

Thank your revision. Material and Methods were changed accordingly to reviewer’s suggestions.

(Discussion) All results of this study are based solely on the SaOS-2 cell line. Therefore, I think the evidence is weak in terms of reproducibility. Please mention that as a limitation in the discussion.

We added the following sentence to the Discussion: “Nevertheless, further investigations on other OS cell lines are necessary to corroborate the mitochondria-mediated mechanism suggested in this study”.

(Overall) -Some abbreviations need to be explained when used for the first time. (e.g., HA, TEM, RT, and PBS.)

Thank you for the careful review. We made explicit the abbreviations where necessary.

Reviewer 3 Report

Osteogenic sarcoma (osteosarcoma) is one of the most common, aggressive, and non-hematologic malignancies that arises from primitive transformed cells of mesenchymal origin. They exhibit osteoblastic differentiation, producing malignant osteoid of the bones. The submission by Rossi and colleagues ‘‘The impact of osteosarcoma cell differentiation on biomineralization: lights and shadows on the mitochondrial involvement in hydroxyapatite formation’’ gives some insights on the Effects of osteosarcoma cell differentiation on biomineralization and the subsequent role of mitochondria in hydroxyapatite formation. Though the manuscript merits publication, there are some points that (will) need to be revised as specified hereunder.

1. Title

At a glance, the phrase ‘‘lights and shadows’’ makes the original research looks like a ‘‘comprehensive’’ review article. I would revise this title to read: Effects of osteosarcoma cell differentiation on biomineralization: Revisiting mitochondrial involvement in hydroxyapatite formation. This is in part because some findings of this study (e.g., L293-294) were already hinted on by previous authors.

2. Keywords: osteosarcoma >> osteogenic sarcoma (since osteosarcoma already appears in the title).

3. Introduction

L38: References [2, 3] should be placed at the end of this paragraph (L39).

L58: Please bring the expansion of HA at L61 to this line.

L70: ER needs to be expanded at first use.

4. Results

L86-89: Please recheck, these clearly describes somehow the methods used.

Apparently, some Supplementary materials (S1, S2 and S3) are missing in this submission.

5. Discussion

L256: days >> day.

L258: Delete ‘’analysis “. It is repetitive because PCA = principal component analysis.

6. Materials and Methods

A significant portion of this submission overlaps previous publication/submitted thesis work (see attached plagiarism report). It would be nice to cite the previous studies from which these methods were excerpted from.  

Some grammatical fixes need to be done. For example, days 10 could be revised to day 10

Author Response

  1. Title

At a glance, the phrase ‘‘lights and shadows’’ makes the original research looks like a ‘‘comprehensive’’ review article. I would revise this title to read: Effects of osteosarcoma cell differentiation on biomineralization: Revisiting mitochondrial involvement in hydroxyapatite formation. This is in part because some findings of this study (e.g., L293-294) were already hinted on by previous authors.

Thank you for your suggestion: we changed the title to “Shedding light on osteosarcoma cell differentiation: impact on biomineralization and mitochondria morphology”. As has been highlighted by the reviewer, the previous title was misleading.

  1. Keywords: osteosarcoma >> osteogenic sarcoma (since osteosarcoma already appears in the title).

Thank you for your suggestion.

  1. Introduction

L38: References [2, 3] should be placed at the end of this paragraph (L39).

L58: Please bring the expansion of HA at L61 to this line.

L70: ER needs to be expanded at first use.

Thank you for the careful review. We moved the references as suggested and made explicit the abbreviations where necessary.

  1. Results

L86-89: Please recheck, these clearly describes somehow the methods used.

Apparently, some Supplementary materials (S1, S2 and S3) are missing in this submission.

L86-89 were deleted from the section as not useful for the purpose of the section.

We apologize for the inconvenience. Supplementary information will be submitted with the manuscript.

  1. Discussion

L256: days >> day.

L258: Delete ‘’analysis “. It is repetitive because PCA = principal component analysis.

Thank you for the careful review. Corrections were made accordingly to reviewer suggestions.

  1. Materials and Methods

A significant portion of this submission overlaps previous publication/submitted thesis work (see attached plagiarism report). It would be nice to cite the previous studies from which these methods were excerpted from. 

Thank you for the revision. The text was changed accordingly and references were added where necessary.

Some grammatical fixes need to be done. For example, days 10 could be revised to day 10

Revisions and changes were done.

Reviewer 4 Report

Francesca Rossi and co-authors present a quality and well-written experimental manuscript focused on the impact of osteosarcoma cell differentiation on biomineralization with regards to the mitochondrial involvement in hydroxyapatite formation.

Authors report here that they induced the osteoblast-like human osteosarcoma cell line SaOS-2 to differentiate into a mature osteoblastic phenotype, which was compared to OS cells used as reference. The early stages of bone biomineralization were studied by monitoring the genesis of the first mineral core deposits and their evolution during the differentiation process using gene expression analysis, histochemical staining, enzyme activity assays and transmission electron microscopy.

Authors used synchrotron-based cryo-soft X-ray tomography (SXT) and cryo-XANES spectro-microscopy to follow the chemical evolution of mineral depositions over time and localize them in the cellular ultrastructure, allowing to simultaneously track morphological variations in intracellular organelles at nanometric resolution.

Finally, authors conclude that this study sheds light on the genesis and evolution of HA in differentiating osteosarcoma cells, identifying HA as the final product of the restored biomineralization. Furthermore, their findings suggest a mechanism for mineral matrix transport based on the interaction between mitochondria and vesicles. Finally, through a morpho logical study of mitochondria, reprogramming of OS cells induced to differentiate was observed, as similarly reported for other cancer types. These striking findings reinforce the theory that osteosarcoma originates from defective osteoblastic differentiation, supporting therapeutic approaches based on differentiating agents that might restore physiological mineralization in OS cells.

==============================

Overall, the manuscript is highly valuable for the scientific community and should be accepted for publication after the corrections are made.

Other comments:

1) Please check for typos throughout the manuscript.

Author Response

Overall, the manuscript is highly valuable for the scientific community and should be accepted for publication after the corrections are made.

Other comments:

1) Please check for typos throughout the manuscript

Thank you for your kind comments. We appreciate the time and the effort that you have dedicated to providing a valuable feedback on our manuscript.

Typos were fully checked in text and corrected where necessary.